# Serum *N*-Glycosylation RPLC-FD-MS Assay to Assess Colorectal Cancer Surgical Interventions

**DOI:** 10.3390/biom13060896

**Published:** 2023-05-27

**Authors:** Alan B. Moran, Georgia Elgood-Hunt, Yuri E. M. van der Burgt, Manfred Wuhrer, Wilma E. Mesker, Rob A. E. M. Tollenaar, Daniel I. R. Spencer, Guinevere S. M. Lageveen-Kammeijer

**Affiliations:** 1Center for Proteomics and Metabolomics, Leiden University Medical Center, 2300 RC Leiden, The Netherlands; 2Ludger Ltd., Culham Science Centre, Abingdon OX14 3EB, UK; 3Department of Surgery, Leiden University Medical Center, 2300 RC Leiden, The Netherlands; 4Department of Analytical Biochemistry, Groningen Research Institute of Pharmacy, University of Groningen, 9713 AV Groningen, The Netherlands

**Keywords:** sialic acid derivatization, reversed-phase liquid chromatography, mass spectrometry, serum, *N*-glycosylation, colorectal cancer, surgery

## Abstract

A newly developed analytical strategy was applied to profile the total serum *N*-glycome of 64 colorectal cancer (CRC) patients before and after surgical intervention. In this cohort, it was previously found that serum *N*-glycome alterations in CRC were associated with patient survival. Here, fluorescent labeling of serum *N*-glycans was applied using procainamide and followed by sialic acid derivatization specific for α2,6- and α2,3-linkage types via ethyl esterification and amidation, respectively. This strategy allowed efficient separation of specific positional isomers on reversed-phase liquid chromatography–fluorescence detection–mass spectrometry (RPLC-FD-MS) and complemented the previous glycomics data based on matrix-assisted laser desorption/ionization (MALDI)-MS that did not include such separations. The results from comparing pre-operative CRC to post-operative samples were in agreement with studies that identified a decrease in di-antennary structures with core fucosylation and an increase in sialylated tri- and tetra-antennary *N*-glycans in CRC patient sera. Pre-operative abundances of *N*-glycans showed good performance for the classification of adenocarcinoma and led to the revisit of the previous MALDI-MS dataset with regard to histological and clinical data. This strategy has the potential to monitor patient profiles before, during, and after clinical events such as treatment, therapy, or surgery and should also be further explored.

## 1. Introduction

Glycans drastically affect the function of a glycoprotein and contribute to the malignant phenotype of cancer cells by promoting proliferation, metastasis, and immunosuppression [1,2,3]. In this regard, aberrant glycosylation profiles add to understanding pathological steps of tumor development and progression and have become a hallmark of cancer [4,5]. For example, in the case of colorectal cancer (CRC), aberrant *N*-glycosylation with regard to the tumor microenvironment is increasingly studied in CRC tissue samples using imaging mass spectrometry (MS) [6,7,8,9]. Other glycomics studies also reported associations of *N*-glycome alterations with survival and tumor stage in CRC [10,11,12]. Overall, there is evidence to support that earlier diagnosis of cancer increases the success of curative treatment and long-term survival in general and has motivated various omics fields to explore retrospectively collected serum samples from cancer patients [13].

Previously we have applied matrix-assisted laser desorption/ionization (MALDI)-MS as a readout for total serum *N*-glycome analysis from CRC patients, where it was found that serum *N*-glycome alterations in CRC were associated with patient survival [14]. Although MALDI-MS is a fast detection technique, it does not include any separation of positional isomers [15]. In contrast, recently we developed a platform whereby plasma *N*-glycans are fluorescently labeled using procainamide followed by sialic acid linkage-specific derivatization via ethyl esterification and amidation, which allows efficient separation of sialylated and positional isomers on reversed-phase liquid chromatography–fluorescence detection–mass spectrometry (RPLC-FD-MS) [16]. The performance of this assay was reported for plasma samples, but this method is also suitable for serum-based specimens. In particular, positional isomers related to galactose occupation on the glycan arm (α3 versus α6), as well as fucose location (antennary versus core), were separated. Thus, we reported this effective analytical strategy by benchmarking the performance of a complex mixture of released plasma *N*-glycans against the gold standard: hydrophilic interaction liquid chromatography (HILIC)-FD-MS [16]. We demonstrated sufficient complementarity between both platforms as an overlap of up to 95% of the total relative area was demonstrated with the HILIC-FD-MS approach. Moreover, the repeatability and intermediate precision of the RPLC-FD-MS platform were determined at a median interday (*n* = 50) relative standard deviation (RSD) of 5.4% for the 10 most abundant *N*-glycans. The overall throughput, compared to a HILIC approach for released *N*-glycan analysis, was improved as the sample preparation protocol is executed on a liquid-handling robot and a 35 min separation gradient is applied.

In the current work, we re-visit the previous MALDI-MS-based serum glycomics study in order to demonstrate the clinical validity of the developed method, [14] and apply the RPLC-FD-MS platform on a subset of the samples, namely pre- and post-operative serum samples obtained from 64 CRC patients. In particular, we aim to uncover relevant serum *N*-glycomic signatures in the context of CRC specifically related to sialic acid linkages as well as positional isomeric structures. The results will be compared with those from the MALDI-MS-based measurements of the pre- and post-operative samples and, furthermore, discussed in the context of histological type and cases versus controls differences [14].

## 2. Materials and Methods

### 2.1. Study Design

Pre- and post-operative serum samples were collected from 64 CRC patients during the period October 2002 and March 2013 using a standardized protocol by the LUMC Surgical Oncology Biobank, resulting in 128 samples in total. Post-operative samples were defined as “cancer-free” based on a post-operative washout period (45 days), absence of recurrence, and a follow-up assessment after at least 1393 days. The cohort characteristics, following data curation steps that are described in the following sections, are shown in Table 1. The procedure for sample collection and storage of pre- and post-operative patient serum samples was previously reported [14]. Serum tubes (BD Vacutainer SST II plastic serum tubes, BD, Plymouth, UK) were used to draw blood before being centrifuged within a maximum of four hours. Both pre-operative and post-operative samples were stored for 30 days or less at −20 °C before biobank collection and were then stored at −80 °C. Aliquoting was carried out using a Microlab STAR line pipetting robot (Hamilton, Bonaduz, Switzerland) into sterile 2D barcoded V-bottom 500 µL tubes (Thermo Scientific, Hudson, NH, USA). Prior to this, samples were thawed on ice and the aliquots were stored at −80 °C. Prior to sample preparation, the samples were stored at −20 °C before being thawed in order to begin sample processing.

The entire workflow, including PNGase-F release, procainamide fluorescent labeling, ethyl esterification and amidation derivatization, and RPLC-FD-MS measurement, has previously been described [16] and is briefly explained in the following sections. The sample preparation protocol was performed using a Hamilton Microlab STARlet liquid-handling robot.

### 2.2. N-Glycan Release and Fluorescent Labeling

The entire cohort of pre- and post-operative serum samples was contained in two plates. Serum (5 µL) from each sample was manually and randomly added to a 96-well PCR plate (4ti-0960/C, 4titude Ltd., Surrey, UK). Importantly, the samples were distributed over two plates in total and the paired pre- and post-operative samples from the same patient were contained on the same 96-well PCR plate. Commercial plasma (P9523, Sigma-Aldrich, Dorset, UK) with a concentration of 1 mg/mL (*n* = 26, 13 per plate) was included as a positive control, deionized water (*n* = 6, 18.2 MΩ resistivity, Sartorius Arium Comfort, Goettingen, Germany) was processed as a negative control on plate two. This was followed by the addition of 4 µL deionized water and 1 µL of a 10× denaturing solution obtained from an *N*-glycan release kit [LZ-rPNGASEF-96, Ludger Ltd., Abingdon, UK). Following this, incubation was performed at 100 °C for 10 min. The plate was removed from the oven and allowed to cool to room temperature before deionized water (6 µL), reaction buffer (2 µL, 500 mM sodium phosphate, pH 7.5), and 10% NP-40 (2 µL) were added to each well. The PNGase F solution was diluted 1:1 (*v*/*v*) with PNGase F storage buffer, which consisted of 50 mM sodium chloride, 5 mM ethylenediaminetetraacetic acid, and 20 mM tris-hydrochloric acid (150 µL, New England Biolabs, Hitchin, UK). Then, 2 µL was added to each sample and the plate was incubated (37 °C) overnight. The next day, a 5% formic acid (Sigma-Aldrich)/deionized water solution (*v*/*v*; 5 µL) was added to each well and incubation was performed at room temperature for 40 min.

A clean-up procedure was performed using a Protein Binding Membrane (PBM) plate [LC-PBM-96, Ludger Ltd.]. The plate was prepared with successive washes of methanol (100 µL) and deionized water (300 µL) followed by the addition of sample to the PBM plate. The PCR plate was washed with deionized water (90 µL), which was also added to the PBM plate. Following this, a 2 mL collection plate [LP-COLLPLATE-2ML-96, Ludger Ltd.] was placed underneath the PBM plate and elution was carried out via centrifugation at 800 rcf (22 °C) for 3 min. An additional 100 µL deionized water was added to each well and another centrifugation step was performed. The samples were transferred to a 300 µL 96-well PCR plate [4ti-0710/C, 4titude Ltd.] and dried down using a vacuum centrifuge.

Procainamide labeling solution [LT-KPROC-96, Ludger Ltd.] was prepared according to the vendor’s guidelines. The labeling solution (20 µL) was added to each well of dried released *N*-glycans and incubated (65 °C) for 60 min. After 10 min, the plate was removed, vortexed, and centrifugated before being placed back in the oven for the remaining incubation time. A HILIC clean-up plate [LC-PROC-96, Ludger Ltd.] was prepared by washing with solutions of 70% ethanol (200 µL, Sigma-Aldrich)/deionized water (*v*/*v*), deionized water (200 µL), and acetonitrile (CH_3_CN, 200 µL, Romil Ltd., Charlton, UK). The sample was added to the plate followed by the addition of CH_3_CN (230 µL) to each well. A vacuum was applied before three further CH_3_CN washes (200 µL) were carried out. The samples were eluted in 100 µL of deionized water into a 96-well 2 mL collection plate [LP-COLLPLATE-2 ML-96, Ludger Ltd.]. Following this, a second elution was carried out using deionized water which resulted in a final volume of 200 µL.

### 2.3. Sialic Acid Derivatization

The derivatization technique has been reported in the literature [17,18] and the protocol was followed as previously described. [16] The released and labeled *N*-glycans were transferred (120 µL) to a 300 µL 96-well PCR plate and dried down using a vacuum centrifuge. Following this, the samples were reconstituted in deionized water (15 µL) and 3 µL was added to 60 µL of ethyl esterification solution, which consisted of 250 mM 1-ethyl-3-(3-(dimethylamino) propyl) carbodiimide (EDC, Fluorochem, Hadfield, UK) and 250 mM hydrate 1-hydroxybenzotriazole (HOBt, Sigma-Aldrich) dissolved in ethanol. Incubation (37 °C) was performed for 60 min before 28% ammonia (12 µL, Sigma-Aldrich) was added to the samples and the plate was incubated for a further 60 min at 37 °C. Following this, the volume in each well was brought up to 300 µL by the addition of 225 µL of CH_3_CN. A clean-up was performed using the HILIC clean-up plate procedure described above and may also be found in the literature [16]. The fluorescently labeled and derivatized *N*-glycans were eluted in 200 µL deionized water and stored at −20 °C until further analysis was performed.

### 2.4. Reversed-Phase Liquid Chromatography–Fluorescence Detection–Mass Spectrometry

A single workup of each sample was performed followed by single injection on the RPLC-MS platform. The measurements for each plate were carried out separately and additional deionized water blanks (*n* = 6) were included during the measurement of plate one. Samples (95 µL) were added up to 100 µL with CH_3_CN so that the final concentration was 5% CH_3_CN (*v*/*v*). A 150 mm × 2.1 mm, ACE excel 2 C18-PFP column (ACE Ltd., Aberdeen, UK) was prepared on an Ultimate 3000 UHPLC system (Thermo Scientific, Hampshire, UK) and a 20 µL injection was performed. The column temperature was set to 60 °C, and the fluorescence detector (λ_ex_ = 310 nm λ_em_ = 370 nm) bulb power was set to ‘high’ whilst the sensitivity was set to eight. A separation gradient was carried out using a flow rate of 0.4 mL/min and 50 mM ammonium formate as solvent A and 10% CH_3_CN, 0.1% formic acid (*v*/*v*) as solvent B. The gradient consisted of the following steps: 0–26.5 min (15%–95% solvent B), 26.5–30.5 min (95% solvent B), 30.5–32.5 min (95%–15% solvent B), 32.5 to 35.1 min (15% solvent B). A representative chromatogram with numbered peaks is shown in Appendix A.

Electrospray ionization–MS was carried out on an amaZon Speed ETD MS (Bruker Daltonics GmbH, Bremen, Germany). The instrument parameters were as follows: ionization mode, positive mode with enhanced resolution scanning; mass range, *m*/*z* 600–1600 and *m*/*z* 900 set as the target mass; capillary voltage, 4.5 kV; source temperature, 250 °C; gas flow, 10 L/min; max accumulation time, 50.00 ms; ion charge control (ICC) target, 200,000. MS/MS spectra were obtained using helium collision-induced dissociation under the following conditions: max accumulation time MS(n): 40.00 ms; ICC target MS(n): 200,000; precursor ion selection: 3 precursors which are released after 0.2 min; MS(n) isolation width: 4 *m*/*z*; scan range selection MS(n), scale to precursor; automatic MS(n) absolute signal threshold: 25,000; automatic MS(n) relative signal threshold: 5%.

### 2.5. Data Processing

The assignment of serum *N*-glycans was performed based on our previous study [16]. Briefly, this was carried out based on exact mass and retention order as well as diagnostic ions obtained in the MS/MS spectra. As a result, *N*-glycans with differently linked sialic acids are mass differentiated following sialic acid derivatization [16]. In addition, isomers with the same mass but different retention time (RT) are denoted as “isomer 1” (iso1) and “isomer 2” (iso2) based on their relative retention order. Notably, several *N*-glycans were not detected in our previous work yet were identified in this study. In this case, the diagnostic fragment ions that support the assignment of these structures are included in Appendix A. *N*-glycan compositions are displayed according to the notation recommended by the Consortium for Functional Glycomics [19]: *N*-acetylglucosamine (N; blue square), mannose (H; green circle), galactose (H; yellow circle), fucose (F; red triangle), *N*-acetylneuraminic acid (S; purple diamond).

Fluorescence detection (FD) data was exported (.txt) using Chromeleon (version 7.2) and imported into HappyTools (version 0.1-beta1, build 190115a) [20] for processing. The processing parameters for this software have previously been reported [16]. Briefly, these parameters were as follows: datapoints (100), baseline function order (1), background window (1), peak edge type (sigma), peak edge value (2), minimum number of calibrants (4), minimum S/N (27). Importantly, baseline correction was applied and parameters for peak calibration and integration are included in Appendix A. Furthermore, a peak integration window (70% FWHM) was implemented, which allowed sufficient peak area quantification while also minimizing peak overlap. In addition, previously optimized quality control parameters [16] were applied in order to perform data curation of fluorescent chromatogram peaks. For example, the maximum RT deviation was ±3 s and a S/N > 9 was required. With regard to the latter, a more lenient S/N criterion in comparison with analytical validation guidelines (S/N > 10) [21] was implemented. Furthermore, peaks eluting at the expected RT were required to be greater than the average intensity plus nine times the standard deviation (s.d.) of the same integration window in the blank and negative control measurements. A single measurement from the analysis did not meet the quality control requirements and, therefore, both paired measurements were removed from the study, resulting in a final sample size of *n* = 64 (128 measurements).

Raw RPLC-MS data acquired in the Bruker data format (.d) were converted into the .mzXML format and imported in LaCyTools (version 2.0, build 200723) [22]. Glycan compositions used for spectra alignment and extraction are included in Appendix A. Further processing parameters used for LaCyTools have previously been described [16]. Importantly, the area for each charge state of each glycan composition was integrated with at least 95% coverage of the theoretical isotopic distribution. Data curation parameters were also applied, including setting the *m*/*z* tolerance to ±100 ppm, and a S/N > 9 was used. Furthermore, the average isotopic pattern quality for each measurement was assessed for the [M + 2H]^2+^, [M + 3H]^3+^, and [M + 4H]^4+^ charge states meeting the above criteria. In this regard, the threshold was set to the average of all charge states plus one s.d. As a result, the isotopic pattern quality threshold was set to a deviation ≤30% from the theoretical isotopic pattern, and this cut-off was applied to assess each charge state separately for every analyte. Finally, curated [M + 2H]^2+^, [M + 3H]^3+^, and [M + 4H]^4+^ ions were individually quantified, followed by summation to provide a total area of all charge states for each glycan followed by adjustment to 100% in order to cover the complete isotopic envelope.

Only fluorescent peaks that contained at least one *N*-glycan that passed the MS curation parameters were considered for further processing. A third quantification approach, termed FD-MS, was determined via the crossover of FD and MS data [16]. This was performed by calculating the local relative proportion of *N*-glycan peaks eluting under the same chromatographic peak based on their MS intensities. Then, the fluorescent signal was multiplied by the proportion of each *N*-glycan eluting underneath that chromatographic peak in order to derive an FD-MS signal for each *N*-glycan composition. Furthermore, the FD-MS data for each *N*-glycan composition were used to calculate “derived glycosylation traits” [14]. In this case, direct *N*-glycan traits refers to individual *N*-glycan compositions, whereas derived traits describe common structural features shared amongst individual *N*-glycans and are based upon the biosynthetic pathway [23]. These include characteristics such as high-mannose, hybrid-type, and complex-type, as well as sialylation, galactosylation, and fucosylation. The formula for each calculated derived trait is included in Appendix A. Importantly, the abundances of isomers of the same *N*-glycan composition were summed in order to calculate derived glycosylation traits.

### 2.6. Statistical Analysis

Principal component analysis (PCA) was performed in order to determine the presence of batch effects within the data. Confounders such as age, sex, and sample plate were explored and the batch effect in all three datasets (FD, MS, FD-MS) due to the sample plate was corrected using linear regression based on the positive control samples [24]. Direct and derived traits were tested for significance based on the results of the Shapiro–Wilk test, a dependent sample *t*-test, or the Wilcoxon signed-rank test in order to determine prognostic capability. A significance threshold (*p*-value < 0.05) with Bonferroni correction was applied. Univariate analysis was performed in Python (Python 3, Create Space, Scotts Valley, CA, USA).

Direct and derived *N*-glycan traits were considered for further analysis when significant differences between pre- and post-operative samples (*p*-value < 0.05 with Bonferroni correction) were observed in the current study, as well as being previously validated by the MALDI-MS study [14]. Associations between those specific direct and derived *N*-glycan traits and disease metadata were explored. This included three categories of the disease, namely stage (1–4), local extent of tumor assessed pathologically (pT1–4), and histological type (Signet-ring cell carcinoma, >50% signet-ring cells; adenocarcinoma, and mucinous adenocarcinoma, >50% mucinous). Thus, to determine whether there was a significant difference in each trait prior to surgery across these categories, a Kruskal–Wallis test was performed with Bonferroni correction, followed by post-hoc analysis looking at the pairwise differences of each subcategory via the Dunn test. To gain an understanding of how well a trait is distinguished by a certain category, multiple models were tested and the model with the highest area under the curve (AUC) is reported while adjusting for age and sex. In this case, the AUC was calculated by comparing one versus the rest, in a 5-fold cross-validation.

## 3. Results & Discussion

*N*-glycomic profiles of pre- and post-operative serum samples from CRC patients were explored using a newly developed workflow (Figure 1) [16]. In this approach, we performed fluorescent procainamide labeling and derivatization of α2,6-(ethyl esterification) and α2,3-(amidation) linked sialic acids followed by RPLC-FD-MS analysis in order to assess serum *N*-glycosylation changes following surgery. In addition to common *N*-glycan features, this method allowed us to perform sialic acid linkage-specific and positional isomer analysis. As shown in Figure 1, we investigated direct *N*-glycan traits as well as derived glycosylation traits the latter of which describes global glycosylation characteristics based on the biosynthetic pathway such as sialylation, galactosylation, bisection, fucosylation, and antennarity. Unless mentioned otherwise, we refer to FD-MS quantification results throughout this study as we previously demonstrated that this technique achieves a more precise relative quantification and greater coverage of glycan structures than solely performing MS or FD quantification [16]. Furthermore, we compared our results with those of a previous study that analyzed the same cohort of patients using a MALDI-MS approach [14]. In this study, we include a comparison of the same 64 patients analyzed by both methods. This allowed us to further validate those results using different ionization, detection, and quantification parameters. Finally, we compared the clinical performance of the three quantification techniques that were applied in our study, namely FD, MS, and the recently described FD-MS quantification approach [16].

### 3.1. Serum N-Glycosylation Analysis of Pre- and Post-Operative CRC Samples

A total of 48 *N*-glycan structures were detected in this study following data curation (Appendix A). Univariate testing was performed in order to assess whether there were any significant differences in the relative abundances of *N*-glycans in pre- and post-operative samples, which resulted in 14 significant *N*-glycans in the FD-MS dataset. Table 2 shows that 20 direct *N*-glycan traits were previously validated by de Vroome et al. using a MALDI-MS approach [14]. Importantly, that study included discovery and validation cohorts using a case/control setup, as well as the same pre- and post-operative CRC cohort that is also assessed by our workflow. From these, 11 *N*-glycans were found in the current study, eight of which were significant discriminators between the pre- and post-operative groups.

The alteration of these eight significant *N*-glycans in CRC is displayed in Figure 2, which shows a lower pre-operative expression of H4N4F1, H5N4F1, H5N4F1S_2,6_1, and H5N4F1S_2,3_2. These results are in line with previous reports that di-antennary structures with core fucosylation are decreased in CRC [14,25,26,27]. In addition, we also observed that H5N5F1 has a lower abundance in pre-operative CRC, which is supported by previous findings whereby bisected *N*-glycan species were also found to be decreased in serum as well as tissues from CRC patients [14,25,28]. In contrast, a higher abundance of larger structures including H6N5F1S_2,3_1S_2,6_1, H6N5F1S_2,3_1S_2,6_2 (isomer 1), and H6N5S_2,6_3 in pre-operative CRC is observed. This is in agreement with results that showed that mainly sialylated tri- and tetra-antennary *N*-glycans were found to be increased in CRC [14,27]. Importantly, the same trend in CRC is observed for these structures by the MALDI-MS approach (Table 2).

We investigated derived glycosylation traits that describe global features of glycosylation based on the biosynthetic pathway using the 18 derived traits previously validated by the MALDI-MS study. In this case, we observed 14 common derived traits, of which 7 were significant differentiators between pre- and post-operative samples (Appendix A). As shown in Figure 3, fucosylated di-antennary glycans (A2F) have a lower expression in pre-operative CRC, which is similar to the results of some of the individual direct *N*-glycan traits mentioned above. In addition, α2,6-sialylated fucosylated glycans (AFE) and α2,6-sialylated di-antennary glycans (A2E) have a greater abundance in pre-operative CRC. In relation to this, it has been reported that α2,6-sialylation is increased in CRC [29]. Moreover, sialylation per antenna across all glycan species (AS) and sialylated di-antennary *N*-glycans (A2S) are increased in pre-operative CRC, possibly due greater abundance of galactosylation per antenna (AG), which would increase the number of available galactose moieties for sialic acid occupation. Furthermore, an increase in α2,3-sialylation is observed specifically in fucosylated tri-antennary glycans (A3FAm) in pre-operative CRC. As shown in Appendix A, this derived trait is composed of H6N5F1S_2,3_1S_2,6_1 and H6N5F1S_2,3_1S_2,6_2 (both isomers summed). Appendix A shows that antennary fucosylation was observed in the case of H6N5F1S_2,3_1S_2,6_1 and H6N5F1S_2,3_1S_2,6_2 (isomer 1). Thus, this observation supports previous findings that CRC is associated with an increase in sialyl-Lewis epitopes [30,31]. Moreover, Rebello et al. analyzed a subset of the same pre- and post-operative cohort using an antennary fucosylation MALDI-MS assay and uncovered that sialyl-Lewis X and Lewis X antigens were significantly increased in pre-operative samples [32]. Thus, our FD-MS results are consistent with those reported by the MALDI-MS [14] approach, indicating the same direction of these seven derived traits in CRC.

The analysis of the serum *N*-glycome in the pre- and post-operative CRC cohort by MALDI-MS previously resulted in the detection of 83 *N*-glycan compositions (Table 2) [14]. In comparison, as previously mentioned, we observed a total of 48 *N*-glycan structures in the current study. Whereas an ion trap was utilized in the current workflow, it is notable that de Vroome et al. employed a Bruker ultrafleXtreme equipped with a TOF mass analyzer operated in reflection mode, which generally demonstrates greater performance in areas such as mass resolution and mass accuracy [33]. For example, the mass resolution of low, medium, and high abundance *N*-glycans measured by the ion trap MS was investigated. In this case, a mass resolution of 6500, 5290, and 6540 was obtained for H4N5 (*m*/*z* 951.40), H6N5S_2,3_1S_2,6_2 (isomer 2, *m*/*z* 1052.43), and H5N4S_2,6_2 (*m*/*z* 834.01), respectively. In relation to this, the isotopic profile of H6N5S_2,3_1S_2,6_2 (isomer 2) is shown in Appendix A. Takei et al. analyzed the serum *N*-glycome using a previous iteration of the Bruker MALDI instrument, the ultraflex III, which resulted in the quantification of 34 *N*-glycans [25]. Thus, it is likely that the current workflow could be improved by coupling with a mass spectrometer with higher resolving power and greater sensitivity. Moreover, further improvements in sensitivity could be gained by increasing the injection amount or implementing a nanoflow column.

### 3.2. Positional Isomers in CRC N-Glycome

Our workflow demonstrated the efficient chromatographic separation of positional isomers using RPLC following fluorescent labeling and sialic acid derivatization [16]. This is evident in Table 2, which shows that seven *N*-glycan compositions with the same *m*/*z* were detected as positional isomers, resulting in the assignment of 14 structures in total. We further investigated the structural components that may lead to different retention times for positional isomeric structures. To perform this, we applied our workflow in order to examine IgG *N*-glycans, for which H4N4F1 (G1F, *m*/*z* 922.89) is a well-known composition that consists of two isomers [34]. As shown by Appendix A, we observed a longer retention time for H4N4F1 with α3-antenna galactose occupation using a 70 min gradient, indicating that structural variations may lead to different retention times for certain positional isomeric *N*-glycans. Despite this, there is a complex interplay of influences on *N*-glycan retention during RPLC separation, including sialic acid-linkage dependent derivatization, galactose occupation by sialic acids, bisection, number of hexose residues, and fucosylation [16]. As a result, the specific assignment of structures remains challenging for *N*-glycans; in particular for those with two antennae or more. Nonetheless, in the case of relevant structures related to a disease, an exoglycosidase approach could be followed in order to characterize specific isomeric features of interest. Following this, this method could then be applied in order to investigate those structures within a clinical context.

In our study, we revealed that specific isomers are significantly associated with CRC. For example, Figure 4(A1–D1) showed four compositions where only one of the isomeric forms was a significant differentiator between the pre- and post-operative groups, including a previously validated composition, H6N5F1S_2,3_1S_2,6_2. Moreover, Figure 4(A2–D2) also shows that there were significant differences between the isomers in CRC fold-change (pre-/post-) for three of these compositions. Previously we showed that antennary fucosylation results in an earlier RT than core fucosylation [16] and, expectedly, Appendix A shows that H6N5F1S_2,3_1S_2,6_2 (isomer 1) eluted earlier than H6N5F1S_2,3_1S_2,6_2 (isomer 2). In addition, the former assignment is supported by the presence of a fragment ion with *m*/*z* 512.20 (Appendix A) which is indicative of antennary fucosylation, whereas the latter showed the presence of a core fucose fragment ion (*m*/*z* 587.33). In contrast, MALDI-MS may only differentiate mass differences between *N*-glycans. In this case, the location of the fucose residue on H6N5F1S_2,3_1S_2,6_2 was not specified by de Vroome et al. [14], whereas it is likely that this composition is a mixture of core and antennary fucosylated forms when detected solely based on *m*/*z*. Evidence for fucose positional isomers has been provided by Rebello et al., who found a reduction in the relative abundance of H6N5F1S_2,3_1S_2,6_2 following core fucosidase digestion [32]. Thus, here we observed both isomer forms of this *N*-glycan and we illustrated that only the antennary fucosylated form (isomer 1) is significantly different between the pre- and post-operative groups. As a result, our workflow allows the assessment of the specific association of antennary fucosylation with CRC.

### 3.3. Investigation of Pre- and Post-Operative Histological Type

A Kruskal–Wallis test was performed in order to determine whether the pre-operative CRC abundances of the direct and derived *N*-glycan traits shown in Figure 2 and Figure 3 were significant across three categories of the disease, namely stage, pT, and histological type. With regard to this, H5N4F1, H4N4F1, and H5N4F1S_2,6_1 demonstrated significant differences for histological type (Figure 5). As shown in Figure 5(A1–C1), it can be seen that differences in the pre-operative abundances of these *N*-glycans are no longer observed following surgery (post-operative). In addition, Figure 5(A2–C2) further demonstrates that the relative abundances converge in the post-operative measurements. Thus, differences in histological type are no longer readily observed following surgery.

In order to further support these results, the *N*-glycans shown in Figure 2 were investigated retrospectively in the MALDI-MS datasets, namely the pre- (*n* = 64 pre-operative cases) and post-operative and discovery and validation studies (*n* = 185 cases). This revealed that the same three *N*-glycans (H5N4F1, H4N4F1, and H5N4F1S_2,6_1) were also significant differentiators of histological type. As shown in Appendix A, a similar trend is observed, whereby differences between the subcategories of histological type are no longer observed following surgery. In addition, Appendix A shows that the results of H5N4F1 and H5N4F1S_2,6_1 were also replicated in cases from the MALDI-MS discovery and validation study. These findings are similar to the results previously demonstrated by de Vroome et al., whereby they showed similar *N*-glycan abundances between the post-operative samples and healthy controls [14], signifying a positive response to surgery.

The potential of the *N*-glycans H5N4F1, H4N4F1, and H5N4F1S_2,6_1 as prognostic markers for distinguishing between different histological types was evaluated using a receiver operating characteristic (ROC) curve. As a result of the small sample size, only the largest subcategory versus a combination of the other subcategories is reported. In the case of histological type, Table 3 displays the classification of adenocarcinoma against signet-ring cell carcinoma and mucinous adenocarcinoma. The analysis of pre- and post-operative CRC samples by the RPLC-FD-MS approach achieved the highest AUC values (ranging from 0.71 to 0.83), and the model combining the three *N*-glycan traits showed the best performance (AUC 0.83). In comparison, the analysis of the same cohort using the MALDI-MS technique demonstrated a moderate performance (AUC 0.76–0.79). Although MALDI-MS showed significant differences for pT, its prognostic performance was poor (AUC 0.50–0.59; Appendix A). Similarly, the analysis of the MALDI-MS discovery and validation study (Appendix A) revealed significant differences yet poor performance for stratifying adenocarcinoma (AUC 0.56–0.60), pT3 (AUC 0.48–0.53), and stage 2 (AUC 0.49). The characteristics of the discovery and validation cohort may be found in Appendix A.

Histological type is an important prognostic factor in CRC, as different types of tumors may show different behavior and response to treatment. For example, signet-ring cell carcinoma is associated with poor differentiation (high-grade) and generally results in a worse outcome than adenocarcinomas [35,36,37,38]. In addition, microsatellite stable mucinous adenocarcinomas may demonstrate more aggressive behavior when detected at an advanced stage [35,39]. Undoubtedly, identifying the histological type of CRC can assist in guiding the treatment approach as well as predicting patient prognosis [35], and yet, interobserver variability can present challenges in this regard [40]. Thus, the results displayed by the current study demonstrate the utility of the serum *N*-glycome in aiding histological type classification. Moreover, we also illustrated that the response to surgery may be monitored by comparing pre- and post-operative *N*-glycan abundances from the same patient, thus promoting an individualized approach to treatment. This is an important result as the need and benefit of personalized medicine are continuously being recognized [41,42]. Despite this, it should be noted that this cohort consisted mainly of the most prominent histological type, adenocarcinoma [43] (Table 1), and as a result, it may be challenging to translate these results to subsequent studies. Nevertheless, this analysis shows the potential of the serum *N*-glycome as a prognostic marker and further validation should be performed, particularly with regard to histological type classification and response to surgery.

### 3.4. Comparison of Clinical Performance of Three Quantification Approaches

In this study, we employed three quantification approaches, FD, MS, and FD-MS, in order to analyze pre- and post-operative CRC. Previously, we showed that FD-MS quantification, due to the detection of specific glycan masses, resulted in greater coverage of glycan features than FD. Furthermore, as a result of incorporating the FD signal into the quantification of individual *N*-glycans, we also demonstrated that FD-MS achieved greater precision than MS quantification [16]. In the current work, we further compared the performance of these quantification approaches in a clinical setting. In Appendix A, the number of significant peaks or glycan structures is provided for the FD (14), MS (11), and FD-MS (14) approaches. In relation to this, the number of validated significant *N*-glycans for MS and FD-MS is seven and eight, respectively. Furthermore, there were seven FD peaks (10, 11, 13, 17, 24, 25, 32) that were significant between the groups and also contained a previously validated *N*-glycan. Thus, it is shown here that the three quantification approaches perform similarly in terms of the number of validated and significant features (FD peak or glycan structure). In addition, the three quantification approaches were also evaluated for their diagnostic performance over time in relation to adenocarcinoma classification via ROC curves, whereby AUCs were also calculated (Table 3). This demonstrates a good performance for FD-MS and FD quantification which both fall into a similar AUC range (AUC 0.70–0.80). In comparison, MS quantification resulted in a fair performance (0.68–0.72). Thus, in comparison with FD, the FD-MS quantification approach showed a similar performance yet allowed the analysis of specific direct *N*-glycan traits in relation to CRC. The comparison of MS and FD-MS demonstrated that the latter showed improved diagnostic performance for adenocarcinoma classification.

Cancer-associated alterations in glycosylation have been well-established in the literature [44,45,46]. Moreover, several studies have demonstrated the potential with regard to the application of glycomics for aiding and improving cancer diagnosis and prognosis [14,47,48]. Despite this, there is still a lack of glycomic tests that have been translated from biomedical research into clinical laboratories [49]. This is partly because of the challenges that carbohydrates themselves present for analytical method development, detection, and quantification, as well as a lack of validation studies beyond the discovery phase [50]. In this study, we have demonstrated that MS detection allows the underlying glycosylation alterations in CRC to be probed. As a result, a greater understanding of the disease may be developed whilst maintaining specificity with regard to the traits that can be targeted for diagnostic and prognostic testing. However, FD of labeled glycans shows promise for clinical development [51], as it may avoid common challenges with glycan analysis by MS, such as poor ionization. In this regard, we have demonstrated similar and satisfactory performance of FD and FD-MS quantification (Table 3). Moreover, FD-MS relative abundances more closely resemble FD relative abundances in comparison with MS-based quantification (Appendix A). Thus, upon identification of the best-performing *N*-glycan traits, the FD-MS results could be more easily bridged with the FD platform, enabling efficient implementation within the clinical setting.

## 4. Perspectives

Undoubtedly, as previously mentioned, the results presented here require further validation with a larger and more diverse cohort of CRC patients in order to evaluate the true prognostic potential of serum *N*-glycosylation. In this regard, the monitoring of patient profiles before, during, and after clinical events such as treatment, therapy, or surgery seems promising and should be further explored. In addition, clinical validation of FD-MS quantification and bridging to FD quantification should also be examined. In this regard, interferences that may affect MS and FD signals, such as different mobile phase conditions across the gradient as well as unidentified fluorescent compounds, should be defined and, if possible, eliminated. Overall, the presented workflow is suitable for future investigations into biomarker discovery, as well as clinical validation and translation.

## 5. Conclusions

In this study, we evaluated the clinical performance of our workflow, which performs *N*-glycan fluorescent labeling and sialic acid derivatization followed by RPLC-FD-MS measurement. Under alternative ionization, detection, and quantification conditions, we further supported previously validated direct *N*-glycan and derived glycosylation traits for differentiating pre-operative CRC in comparison with post-operative samples from the same patient. Furthermore, we observed that specific positional isomers were significant between the two groups, which has important clinical implications in the context of this disease. In addition, pre-operative abundances of *N*-glycans showed good performance for the classification of adenocarcinoma, and new clinical insights were gained as we illustrated that differences between histological types were abolished following surgery. Thus, serum *N*-glycosylation is a promising CRC prognostic marker that may be further investigated for performing disease classification and monitoring patient response to surgery.

## Figures and Tables

**Figure 1 biomolecules-13-00896-f001:**
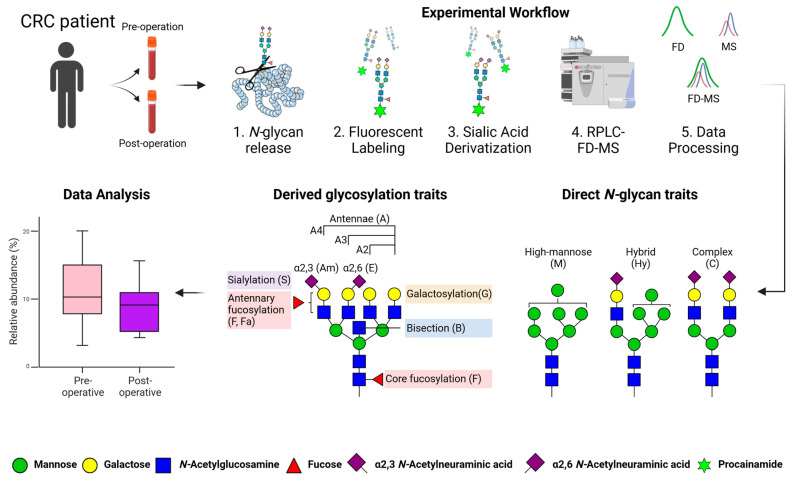
Analysis of pre- and post-operative *N*-glycans in CRC by RPLC-FD-MS. Serum samples were taken pre- and post-operation from the same patient. Each sample underwent fluorescent labeling and sialic acid derivatization *N*-glycan workflow followed by RPLC-FD-MS measurement. Three quantification approaches were used for data processing, namely FD, MS, and a combination of these two approaches via FD-MS. Direct *N*-glycan and derived glycosylation traits were determined in order to perform data analysis and comparison of the two groups. Adapted with permission from Moran, A.B., et al., 2022. Analytical Chemistry, 94(18), pp. 663–6648 [16]. Copyright 2022 American Chemical Society.

**Figure 2 biomolecules-13-00896-f002:**
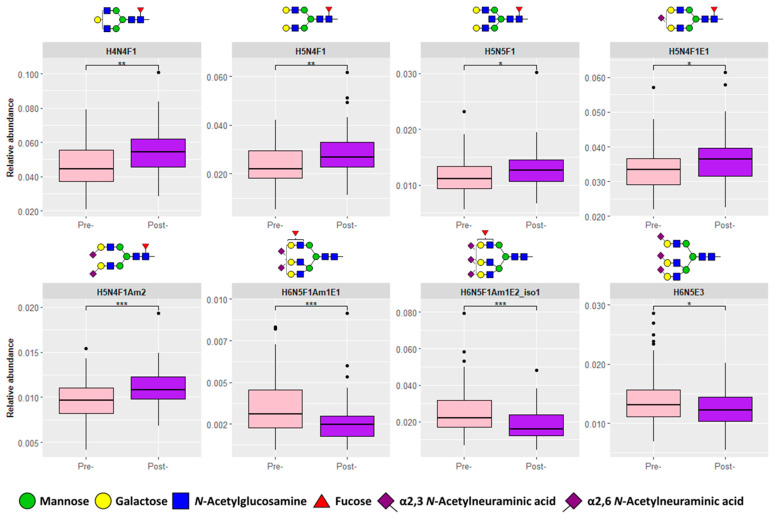
Significant direct *N*-glycan traits. Eight significant direct *N*-glycan traits were found between pre-operative (pre-, *n* = 64) and post-operative (post-, *n* = 64) samples that were also previously validated [14]: H4N4F1, H5N4F1, H5N5F1, H5N4F1S_2,6_1, H5N4F1S_2,3_2, H6N5F1S_2,3_1S_2,6_1, H6N5F1S_2,3_1S_2,6_2 (isomer 1), and H6N5S_2,6_3. Asterisks denote significance with *p*-value < 0.05 (*), 0.01 (**), 0.001 (***). Abbreviations: hexose (H), *N*-acetylhexosamine (N), fucose (F), amidated α2,3-linked *N*-acetylneuraminic acid (Am), ethyl esterified α2,6-linked *N*-acetylneuraminic acid (E), and isomer (iso).

**Figure 3 biomolecules-13-00896-f003:**
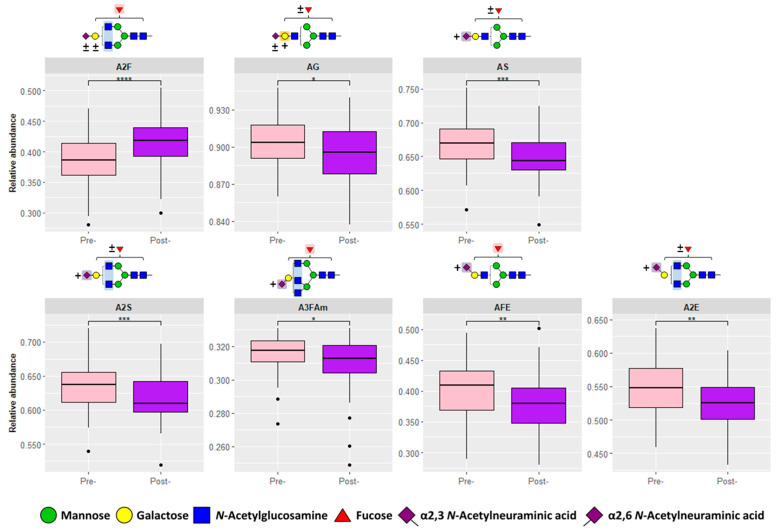
Significant derived glycosylation traits. Seven significant derived glycosylation traits were found between pre-operative (pre-, *n* = 64) and post-operative (post-, *n* = 64) samples that were also previously validated [14]: A2F (fucosylated diantennary glycans), AG (galactosylation per antenna), AS (sialic acid per antenna), A2S (sialic acids per diantennary glycan), A3FAm (α2,3-linked sialic acids per fucosylated triantennary glycan), AFE (α2,6-linked sialic acids per fucosylated glycan), and A2E (α2,6-linked sialic acids per diantennary glycan). Asterisks denote significance with *p*-value < 0.05 (*), 0.01 (**), 0.001 (***), 0.0001 (****). Monosaccharide ranges are specified via plus-minus (±; 0–4) and plus (+; 1–4) symbols. Abbreviations: antennae (A), fucose (F), galactosylation (G), *N*-acetylneuraminic acid (S), amidated α2,3-linked *N*-acetylneuraminic acid (Am), and ethyl esterified α2,6-linked *N*-acetylneuraminic acid (E).

**Figure 4 biomolecules-13-00896-f004:**
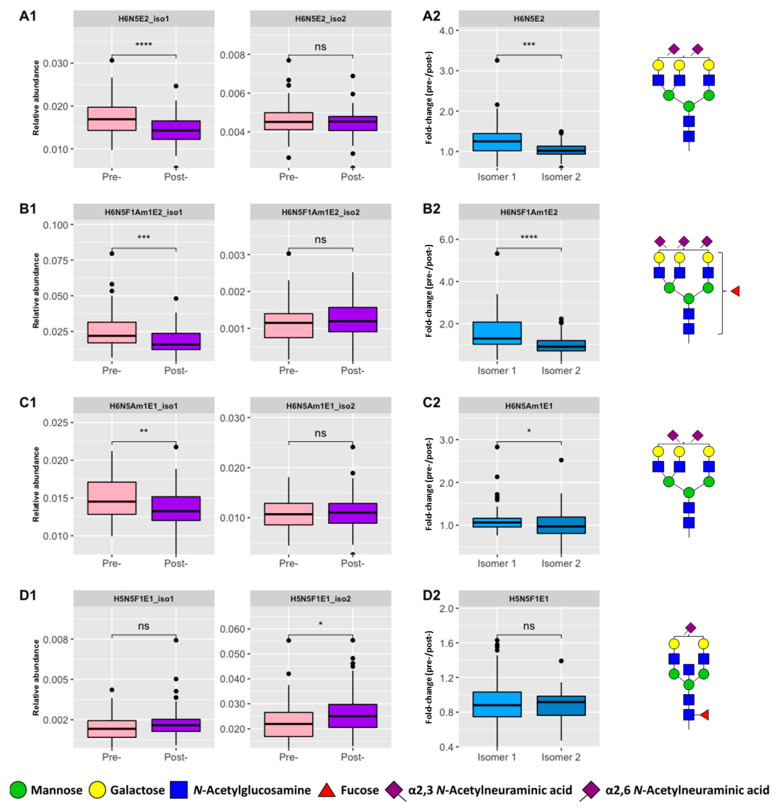
Positional isomeric *N*-glycans that show significant differences in CRC. Four positional isomers with significant differences were observed, as denoted by each row. (**A1**–**D1**) Boxplots of relative abundances observed in pre- (*n* = 64) and post-operative (*n* = 64) samples for each isomeric form. (**A2**–**D2**) Fold-change in CRC (pre-/post-, *n* = 64) for each isomer whereby an increase (>1) or a decrease (<1) may be observed. Asterisks denote significance with *p*-value < 0.05 (*), 0.01 (**), 0.001 (***), 0.0001 (****). Abbreviations: Not significant (ns), hexose (H), *N*-acetylhexosamine (N), fucose (F), amidated α2,3-linked *N*-acetylneuraminic acid (Am), and ethyl esterified α2,6-linked *N*-acetylneuraminic acid (E).

**Figure 5 biomolecules-13-00896-f005:**
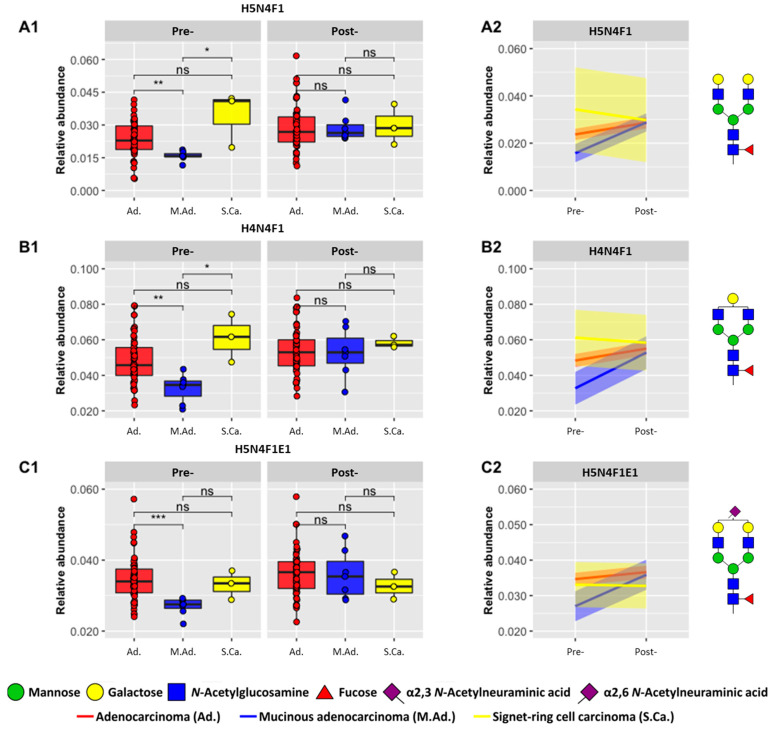
Differences in histological type between pre- and post-operative CRC samples. Three *N*-glycans differentiated significantly based upon histological type, adenocarcinoma (*n =* 57), mucinous adenocarcinoma (*n* = 7), and signet-ring cell carcinoma (*n* = 3), as denoted by each row: row A (H5N4F1), row B (H4N4F1) and row C (H5N4F1S_2,6_1). (**A1**–**C1**) Boxplots of relative abundances observed in pre- and post-operative samples. (**A2**–**C2**) Trend observed following surgery, from pre- to post-operative samples from the same patients. The 95% CI is shown as colored bands around each line. Asterisks denote significance with *p*-value < 0.05 (*), 0.01 (**), 0.001 (***). Abbreviations: Not significant (ns), hexose (H), *N*-acetylhexosamine (N), fucose (F), amidated α2,3-linked *N*-acetylneuraminic acid (Am), and ethyl esterified α2,6-linked *N*-acetylneuraminic acid €.

**Table 1 biomolecules-13-00896-t001:** Clinical characteristics of the pre- and post-operative CRC cohort. Samples were taken before and after surgery from the same patient. A total of 64 patients were included in the analysis following data curation. Two measurements per patient were carried out in order to process the pre- and post-operative samples. Full metadata were collected for each patient in three categories of the disease: stage (1–4), local extent of tumor assessed pathologically (pT1–4), and histological type (Signet-ring cell carcinoma, >50% signet-ring cells; adenocarcinoma, and mucinous adenocarcinoma, >50% mucinous). Standard deviation (s.d.).

Pre- and Post-Operative CRC Cohort (*n* = 64)		
Female sex, *n* (%)		33 (51.6)
Age in years, mean (s.d.)		64.2 (12.7)
Stage, *n* (%)	1	13 (20.3)
	2	31 (48.4)
	3	16 (25.0)
	4	4 (6.3)
pT, *n* (%)	pT1	7 (10.9)
	pT2	8 (12.5)
	pT3	42 (65.6)
	pT4	7 (10.9)
Histological type, *n* (%)	Signet-ring cell carcinoma	3 (4.7)
	Adenocarcinoma	54 (84.4)
	Mucinous adenocarcinoma	7 (10.9)

**Table 2 biomolecules-13-00896-t002:** Comparison of RPLC-FD-MS and MALDI-MS results. Asterisk (*) denotes significant *N*-glycans that were found by the RPLC-FD-MS approach. CRC alteration was calculated as the pre-operative relative abundance divided by the post-operative relative abundance. All results may be found in Appendix A. Abbreviations: n/a: not applicable.

Approach	All *N*-Glycans					Validated *N*-Glycans				
	Count			CRC Alteration		Count		CRC Alteration		
	Total	Common	Positional Isomers	Same	Other	Total	Common	Significant *		
								#	Same	Other
RPLC-FD-MS	48	39	14	30 (77%)	9 (23%)	n/a	11	8	8 (100%)	0 (0%)
MALDI-MS	83		0			20		n/a		

**Table 3 biomolecules-13-00896-t003:** Comparison of AUCs for adenocarcinoma classification in pre-/post-operative CRC. AUCs from the comparison of adenocarcinoma (*n* = 57) versus signet-ring cell carcinoma (*n* = 3) and mucinous adenocarcinoma (*n* = 7) are shown. Three quantification approaches performed by this study (RPLC-FD, RPLC-MS, RPLC-FD-MS) are displayed, as well as data from a previous study performing MALDI-MS [14]. “Combination” refers to the combination of the significant features for histological type. All results may be found in Appendix A. Abbreviations: Not detected (n.d.).

Peak	RPLC-FD	Direct *N*-Glycan Traits	RPLC-MS	RPLC-FD-MS	MALDI-MS
	AUC		AUC	AUC	AUC
10	0.79	H5N4F1	0.70	0.79	0.79
		H4N4F1	0.72	0.71	0.77
n.d.	n.d.	H5N4F1S_2,6_1	0.70	0.80	0.76
n.d.	n.d.	Combination	0.68	0.83	0.77

## Data Availability

The raw data is available from the authors upon request.

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
