# Peer review of "Serum N-Glycosylation RPLC-FD-MS Assay to Assess Colorectal Cancer Surgical Interventions"

_biomolecules, 2023, doi:10.3390/biom13060896_

Round 1

Reviewer 2 Report

The article is well written and tecnically sound. The progress with respect of previous data, including those from the same authors, is limited to the methodological procedure. The relevance by the biomedical point of view is poor, and this may affect the interest to the readers. I have no suggestions for immediate improvement of the article, the  question raises as to wether it is appropriate for the Journal "Biomolecules".

Reviewer 3 Report

In the manuscript titled “Serum N-glycosylation RPLC-FD-MS Assay to Assess Colorectal Cancer Surgical Interventions” by Moran et al. the authors have developed and applied a method using RPLC-FD-MS to profile the N-glycans of CRC patients before and after surgical intervention and identifying glycans that differ in before and after surgery cohorts. Authors have developed a method where the plasma N-glycans are fluorescently labeled using procainamide followed by sialic acid linkage-specific derivatization, and then analyzed them using RPLC-FC-MS. Firstly, the authors have demonstrated the separation of the positional isomers of N-glycans. Secondly, the results were also compared to a previous MALD-MS study. And thirdly, the authors compared the performance of quantification obtained by FD, FD-MS and MS-MS. This is an ambitious effort by the authors. And there is merit to such a study indeed- helping to identify potential prognostic biomarkers. But it appears that the authors have tried to pack in a lot of information into one manuscript in an effort to create a star study. To me it appears like a data deluge and the manuscript does get a bit challenging to navigate in places. It seems obvious as to why the authors wanted to compare their study to the MALDI study. Authors may consider splitting the manuscript into two manuscripts and have them published concurrently. This may also allow the authors to present some of the mass spectrometry data in the main text.

I have some specific comments below. 

Specific Comments:

1.     Authors should correct the number of CRC samples finally used in the study to 64 even though they started with 65 samples.

2.     On what basis did the authors decide to focus the study on Neu5Ac isomers only? Is there evidence in literature to support their being implicated in CRC- biomarker?

3.     I am curious to know the rationale for choosing to perform ETD?

4.     How did the authors decide on certain metrics like max deviation in RT to +/- 3 sec on RT and a S/N > 9 & 30% threshold?

5.     Can the authors clarify how the isomers were determined in FD, FD-MS and MS data sets? I suppose I am trying to understand where the isomers were resolved by RPLC only or also by MS/MS?

6.     Authors have defined what derived traits are, I think that they should also define what direct traits are, to help the readers?

7.     Can authors elaborate on what they mean by significant differences between N-glycans in line 248-249- “ were considered for further analysis when significant differences between pre- and post-operative samples were observed in the current study, as well as being previously validated by the MALDI-MS study”?

8.     I am a bit confused about the labeling method of the (2-6) and (2-3) sialic acids- differing label agent was used for (2-6) vs (2-3) – how did the authors ascertain that the labeling was efficient? Is this a typical method?

9. I suggest that authors should adhere to anyone naming system of the following for consistency – for example the (2-3) isomer is named Am and (2-6) is named E, however in text in lines 316, 322 the nomenclature has been changed. This can be confusing.

10. Do the authors have a handle on why antennary fucosylated glycans have earlier RTs than core ones? Also how are the authors sure that fucose migration has not occurred?

11. Assignment of core vs antennary fucosylated glycan has been based on the presence of a m/z 512.20 fragment vs 587.33 ion – authors should cite relevant study for using such metrics

Round 2

Reviewer 1 Report

The authors have addressed my concerns.

Author Response

Thanks for your review.

Reviewer 3 Report

The revised version of the manuscript with the changes are satisfactory & I recommend the manuscript for publication. 

Author Response

Thanks for your review.